# Text-Adaptive Multiple Visual Prototype Matching for Video-Text Retrieval

**Chengzhi Lin[1], Ancong Wu[1]\*, Junwei Liang[2]**
**Jun Zhang[3], Wenhang Ge[1], Wei-Shi Zheng[1,4,5], Chunhua Shen[6]**
[1]School of Computer Science and Engineering, Sun Yat-sen University, China
[2]AI Thrust, Hong Kong University of Science and Technology (Guangzhou)
[3]Tencent Youtu Lab
[4]Guangdong Province Key Laboratory of Information Security, PR China
[5]Key Laboratory of Machine Intelligence and Advanced Computing, Ministry of Education, China
[6]Zhejiang University
Email: linchzh3@mail2.sysu.edu.cn, wuanc@mail.sysu.edu.cn, junweiliang@hkust-gz.edu.cn,
bobbyjzhang@tencent.com, gewh@mail2.sysu.edu.cn, wszheng@ieee.org, chunhua@me.com

## Abstract

Cross-modal retrieval between videos and texts has gained increasing research interest due to the rapid emergence of videos on the web. Generally, a video contains rich instance and event information and the query text only describes a part of the information. Thus, a video can correspond to multiple different text descriptions and queries. We call this phenomenon the "Video-Text Correspondence Ambiguity" problem. Current techniques mostly concentrate on mining local or multi-level alignment between contents of a video and text (*e.g.*, object to entity and action to verb). It is difficult for these methods to alleviate the video-text correspondence ambiguity by describing a video using only one single feature, which is required to be matched with multiple different text features at the same time. To address this problem, we propose a Text-Adaptive Multiple Visual Prototype Matching model, which automatically captures multiple prototypes to describe a video by adaptive aggregation of video token features. Given a query text, the similarity is determined by the most similar prototype to find correspondence in the video, which is termed text-adaptive matching. To learn diverse prototypes for representing the rich information in videos, we propose a variance loss to encourage different prototypes to attend to different contents of the video. Our method outperforms the state-of-the-art methods on four public video retrieval datasets.

## 1 Introduction

Cross-modal retrieval has gained popularity as a means of matching semantically similar samples across multiple modalities [1, 2, 3, 4]. With the rapid increase in the number of videos, Text-Video Retrieval (TVR) has garnered a lot of interest. Given a text description/video, we aim to retrieve the corresponding video/text description in the gallery. TVR remains a challenging task since the video and text have different semantical descriptions which results in modality gap.

To overcome the modality gap issue, most existing methods generally follow a dual encoding network paradigm which encodes video and text through a visual encoder and a language encoder into the same embedding space, respectively [1, 2, 3, 5, 6]. And then they focus on learning a similarity function between the representations of text and video, which provides the basis for ranking the gallery video/text. However, the modality gap issue is still not well solved due to the misalignment.

---

\*Corresponding author

36th Conference on Neural Information Processing Systems (NeurIPS 2022).

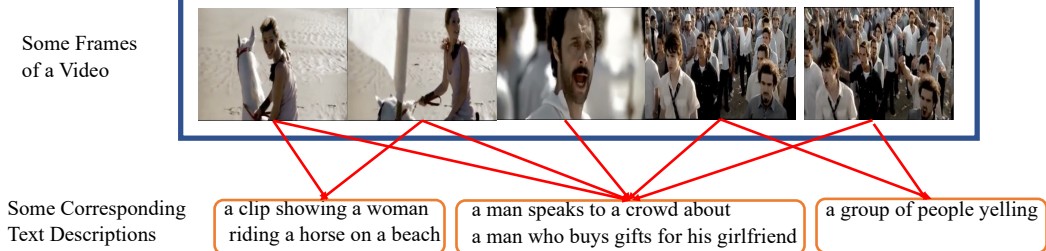

Some Frames of a Video

Some Corresponding Text Descriptions

a clip showing a woman riding a horse on a beach

a man speaks to a crowd about a man who buys gifts for his girlfriend

a group of people yelling

**Figure 1:** An example of text-to-video retrieval. A video can be described by different texts, because different persons focus on different characters when describing the videos. In this paper, multiple video prototypes matching is utilized to help mitigate this issue.

Some methods aim to extract additional information from the video to get more fine-grained video features to be aligned with corresponding text features [5, 7, 8]. They focus on properly combining different video modalities, such as appearance, audio, speech, etc. Some methods pay attention to multi-level alignment, including text entity to video object, text verb to video action, and text phrase to video event [6, 9]. However, most of them ignore the fact that a video contains rich instance and event information, while the query text only describes a part of the information, which leads to video-text correspondence ambiguity problem. Concretely, as shown in Figure 1, due to the multiple characters in the video, the video has different text descriptions: "a woman does something", "a man does something" or "a group does something". The text descriptions are very diverse and scattered in the feature space. With diverse video contents , it is infeasible to align a single video feature with the diverse text features. Neither additional information-based methods nor multi-level alignment-based methods consider such correspondence ambiguity since both of them match a single video feature with multiple text features.

In this work, we focus on video-text retrieval problem and aim to mine fine-grained and diverse event descriptions in a video and adaptively match them with multiple text descriptions to mitigate correspondence ambiguity issue for better alignment. Specifically, we automatically capture multiple prototypes for each video to account for its rich instance and event information by aggregating on token features. Based on multiple prototypes of a video, we aim at finding many-to-many correspondence between prototypes and text representations to reduce ambiguity. To this end, we propose a Text-Adaptive Multiple Visual Prototype Matching Model. We follow the dual encoding network paradigm as previous methods [3]. First, we input video into visual encoder to get the video token features. Then we employ a token-adaptive aggregation method to generate several visual prototypes from the final video token features. As shown in Figure 2, one prototype is the class token and the others are the linear combinations of all the token features. The text feature encoded by the language encoder is matched with the most similar video prototype. This text-adaptive matching can help find text-to-prototype correspondence in video and alleviate the correspondence ambiguity. To further force the prototypes to represent rich information in the video, we propose a variance loss to enhance the diversity of prototypes to attend to different contents of the video.

Extensive experiments were conducted on the MSR-VTT [10], DIDEMO [11], MSVD [12], and LSMDC [13] to evaluate video-text bi-directional retrieval performance. Experimental results show that our proposed model can achieve state-of-the-art results on all the above datasets. Ablation studies were carried out to evaluate the effectiveness of each single part of our model.

The main contributions of our work are three folds. 1) We propose to automatically generate multiple prototypes for video to account for its rich information and introduce a text-adaptive matching strategy to adaptively find correspondence between texts and prototypes. 2) We further design a variance loss to effectively encode diverse information into prototypes to attend to different contents in the video and boost alignment. 3) Our proposed model can achieve new state-of-the-art results on four benchmark datasets. Extensive experiments demonstrate its effectiveness and generalization.

## 2   Related Work

**Text-Video Retrieval.** The methods for text-video retrieval can be classified into three categories: pre-extracted feature based [8, 7, 6, 14, 15, 16], end-to-end training [3, 2] and adaptive matching

methods. Given the small size of most video-text retrieval datasets, the dominant paradigm for video retrieval has been to combine pre-extracted features from "expert" models, such as models trained for a variety of tasks and on multiple modalities such as face, scene, and object recognition, action classification, and sound classification. CE [8] and MMT [5] compute the overall similarity for a video-text pair obtained as a weighted sum of each expert's similarity with the text. DMM [7] uses hierarchical video-text alignment to explore the diverse properties in multi-modal cues for fine-grained alignment. Another important paradigm is to first train on a large-scale vision-and-language dataset and then finetune on downstream dataset. ClipBERT [2] proposes an efficient end-to-end technique based on sparse sampling and demonstrates that the pretrained model by image-language dataset enables superior video-text retrieval initialization. Frozen [3] proposes an end-to-end trainable model that can benefit from large-scale image and video captioning datasets. Some works focus on the matching of different modalities. PCME [4] represents the samples from different modalities as probabilistic distributions in the common embedding space to capture one-to-many correspondences. SSVR [17] highlights the issue of one-to-one assumption in evaluation of video retrieval methods and proposes a new evaluation metric to address this. MQVR [18] tackles the issue of imperfect annotations in existing video retrieval datasets by focusing on the less-studied setting of multi-query video retrieval. Most of them, however, ignore the fact that a video contains rich instance and event information, whereas the query text only describes a portion of the information, resulting in video-text correspondence ambiguity. We propose a Text-Adaptive Multiple Visual Prototype Matching Model to help mitigate this issue.

**Vision Transformer.** Now vision transformer becomes the visual basic backbone in most of the state-of-the-are methods for vision-language tasks [19, 2, 3, 1]. BLIP [1] makes effective use of noisy web data by bootstrapping captions, in which a captioner generates synthetic captions and a filter removes the noisy ones. The work of Frozen [1] makes a minor modification to the Divided Space-Time attention introduced by [20]. Our model is based on the dual framework of Frozen.

# 3 Our Method

A video, in general, contains a lot of rich instance and event information, and the text description only describes a portion of it. A video can have many characters/events/actions, whereas a text typically only describes one character, action, or event. So mining representation of multiple essential events facilitates the text-video matching by reducing the modality gap properly. As a result, a model that generates multiple visual prototypes is better than that generates one feature. With this rationale in mind , we propose our Text-Adaptive Multiple Visual Prototype Matching Model. To generate several visual prototypes from video, we use a token-adaptive aggregation method on the final video token features. Given a query text, the most similar prototype is used to find correspondence in the video, a process called as text-adaptive matching. We also propose a variance loss to encourage different prototypes to attend to different contents of the video in order to learn diverse prototypes for representing the rich information in videos. We define the problem in Section 3.1, present Text-Adaptive Multiple Visual Prototype Matching Model in Section 3.2, describe our variance loss in Section 3.3, and explain our dual encoder in Section 3.4.

## 3.1 Problem Definition

For the video-text retrieval task, we are given $M$ videos $V = \{V_i\}_{i=0}^{M-1}$ and $N$ text descriptions $T = \{T_i\}_{i=0}^{N-1}$. The goal of this task is to learn a similarity function $S : (T, V) \to \mathbf{R}$ . It should maximize positive cross-modal sample similarity while minimizing negative cross-modal sample similarity. In general, we fed video and text to into an vision encoder and language encoder to get normalized feature $f^v, f^v$. Then the similarity function $S$ is defined by

$$S_{t,v} = \langle f^t, f^v \rangle. \tag{1}$$

The similarity of text-video is calculated as the inner production of $f^v, f^t$.

## 3.2 Text-Adaptive Multiple Visual Prototype Matching Model

The similarity function of Equation 1 ignores the Video-Text Correspondence Ambiguity problem: some videos have multiple different text descriptions and queries. To furthermore illustrate this problem, we show the text descriptions diversity of videos by showing the inter-text similarity and

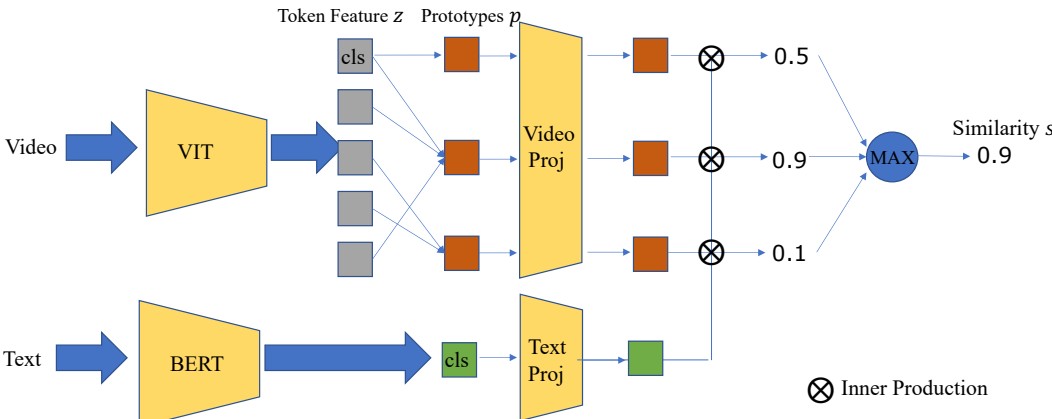

**Figure 2:** Workflow of our method. We feed video into vision transformer to get the token features. Then we employ a token-adaptive aggregation method to generate several visual prototypes from the final video token features. The text feature from the language model is matched with the video prototype which is most similar to text after the projection layer.

the minimum of intra-similarity distribution in 3. We define inter-text similarity as the similarity of different text descriptions for different videos, intra-text similarity as the similarity of different text descriptions for same video. The less the minimum of intra-text similarity, the diversity of the text descriptions for a video. As shown in Figure 3, at least one-third of the minimum of intra-text similarity is less than the mean of inter-text similarity on features from both models. It demonstrates that many videos have diverse captions and contents. So a single feature in some videos cannot be aligned with multiple text features at the same time. As a result, using multiple visual prototypes to align with different text is critical.

An intuitive idea for producing multiple visual prototypes is to generate a feature for each frame/clip of video as a prototype. However, one frame/clip can contain multiple characters and events at the same time. So a better way is to generate many potential prototypes for each frame, then we aggregate them together into multiple visual prototypes. Vision Transformer [3, 2] have shown it's effectiveness on learning multi-modality shared representation. So we take the final layer token features $z \in \mathbf{R}^{B \times D}$ of vision transformer as the potential prototypes. $B, D$ represents the number of tokens, the dimension of token feature.

There are many methods to aggregate the token features to produce multiple visual prototypes, such as transformer decoder, graph neural network, LSTM etc. However, we find that the mask-based production of the prototypes is good enough. The $k$th prototype

$$p_k = \sum_{j=1}^{B} z^j \cdot f^{mask}(z^j)_k, \tag{2}$$

where $z^j, f^{mask}$ represents the jth token feature of $z$, mask-generating function. $f^{mask}$ is implemented by a linear layer, followed by relu function. In general, the class token feature is regarded as the final output of vision transformer [3, 2]. So we expand $p_{k+1} = z^{cls}$, which means we have $K + 1$ visual prototypes. For comparing visual prototypes and text feature on same embedding space, we input $p$ to a projection linear layer, do normalization and get multiple vision features $f^{ve} \in \mathbf{R}^{(K+1) \times D}$. Matching text to video becomes a matter of matching one text feature to multiple visual prototypes. The text is aligned with the video if one of the prototypes is very close to it. Based on this text-adaptive matching, we have new similarity function as

$$S_{t,v}^{\text{TMVM}} = \max_{k=1}^{K+1} \langle f^t, f_k^{ve} \rangle. \tag{3}$$

### 3.3 Variance Loss

Define mask $m_{i,k}^j = f^{mask}(z^j)_{i,k}$, which represents the mask weight on $j$th token from $k$th prototype of $i$th sample . It's an intuitive idea to different prototypes of same video to attend to different contents and the masks have different attention on different tokens. However, as shown in Figure 3,in some

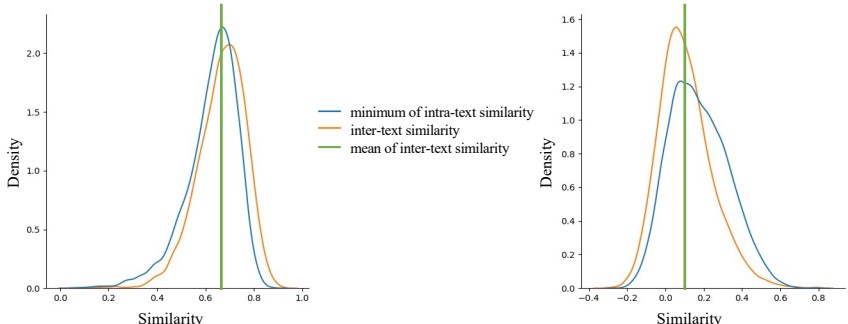

**Figure 3:** Inter-text similarity and minimum of intra-text similarity distribution on MSRVTT sample features. The features of the the left and right graph come from CLIP [19] and SentenceBertFormer [22] respectively. Intra-text similarity is defined as the similarity of different text descriptions for same video. Inter-text similarity is defined as the similarity of different text descriptions for different videos. As a result, the diversity of video contents can be represented by the minimum of intra-text similarity. It can be seen that at least one-third of the minimum of intra-text similarity is less than the mean of inter-text similarity on both graphs. It demonstrates that many videos have diverse captions and contents and it is necessary to use multiple visual prototypes to align with different text descriptions.

videos, the text descriptions are very similar. The videos may be too short and only have a character, event or action. The prototypes and masks are similar . Instead, we'd like all masks of all videos to pay attention to different token positions. Inspired by the variance loss on feature of [21], we design the variance loss on mask as

$$\mathcal{L}_{var} = \frac{1}{B}\sum_{j=1}^{B}\max(0, \gamma - D(m^j, \epsilon)), \tag{4}$$

where $D(\cdot, \cdot)$ is the regularized standard deviation defined by

$$D(x, \epsilon) = \sqrt{\mathrm{Var}(x) + \epsilon}, \tag{5}$$

where $\gamma$ is a constant target value for standard deviation; $\epsilon$ is a small scalar to prevent numerical instabilities. The variance loss forces the variance of all prototype mask values for each token to be large. It also encourages diverse prototypes for representing the rich information in videos.

### 3.4 The Dual-Encoder Framework

We use a end-to-end training dual-encoder framework. Frozen [3], a basic and successful dual-encoder architecture (separate visual encoder and text encoder), serves as our foundation in this study. It's designed for video-language pretraining to match raw-pixel video and raw text with a contrastive loss in dual-encoder framework. The language and vision encoder is multi-layer bidirectional transformer and Timesformer [20] with some changes.

To train this dual-encoder framework, the matched text-video pairings in the batch are regarded as positive, while all other pairwise combinations in the batch are deemed negative. Each video appears at most once in each batch to prevent duplicates. With $L$ text-video pairs in a batch, the symmetrical contrastive loss is calculated as follows

$$\mathcal{L}_{t,v} = \frac{1}{L}\sum_{i}^{L} -\log\frac{\exp(s_{ii}/\tau)}{\sum_{j=1}^{L}\exp(s_{ij}/\tau)} - \log\frac{\exp(s_{ii}/\tau)}{\sum_{j=1}^{L}\exp(s_{ji}/\tau)} \tag{6}$$

where $\tau$ is the temperature and $s_{ii}, s_{ij}$ is the similarity of $i$th paired text-video, $i$th text to $j$th video. Then the ranking of videos/texts is determined by this similarity calculation for single text/video query. So overall loss is defined by

$$\mathcal{L} = \mathcal{L}_{t,v} + \alpha \cdot \mathcal{L}_{var} \tag{7}$$

where $\alpha$ controls the weight of $\mathcal{L}_{var}$.

**Table 1:** Retrieval performance on the MSR-VTT dataset

| Method | Text-to-Video | | | | Video-to-Text | | | | SumR |
|---|---|---|---|---|---|---|---|---|---|
| | R@1 | R@5 | R@10 | MedR | R@1 | R@5 | R@10 | MedR | |
| ActBERT [23] | 8.6 | 23.4 | 33.1 | 36.0 | - | - | - | - | - |
| HowTo100M [14] | 14.9 | 40.2 | 52.8 | 9.0 | 16.8 | 41.7 | 55.1 | 8.0 | 221.5 |
| NoiseEstimation [24] | 17.4 | 41.6 | 53.6 | 8.0 | - | - | - | - | - |
| UviVL [25] | 21.2 | 49.6 | 63.1 | 6.0 | - | - | - | - | - |
| AVLnet [26] | 27.1 | 55.6 | 66.6 | 4.0 | 28.5 | 54.6 | 65.2 | 4.0 | 297.5 |
| HGR [27] | 21.7 | 47.4 | 61.1 | 6.0 | 20.4 | 47.9 | 60.6 | 6.0 | 259.1 |
| TT-CE+ [28] | 29.6 | 61.6 | 74.2 | 3.0 | - | - | - | - | - |
| Dual Encoding [29] | 21.1 | 48.7 | 60.2 | 6.0 | 21.7 | 49.4 | 61.6 | 6.0 | 262.7 |
| TACO [9] | 24.5 | 52.8 | 65.5 | 5.0 | - | - | - | - | - |
| SUPPORT_SET [30] | 30.1 | 58.5 | 69.3 | 3.0 | 28.5 | 58.6 | 71.6 | 3.0 | 316.6 |
| HiT [6] | 30.7 | 60.9 | 73.2 | **2.6** | 32.1 | 62.7 | 74.1 | 3.0 | 333.7 |
| CE [8] | 20.9 | 48.8 | 62.4 | 6.0 | 20.6 | 50.3 | 64.0 | 5.3 | 267.0 |
| MMT [5] | 26.6 | 57.1 | 69.6 | 4.0 | 27.0 | 57.5 | 69.7 | 3.7 | 307.5 |
| DMM [7] | 31.2 | 62.8 | **76.4** | 3.0 | 29.4 | 63.4 | **75.0** | **3.0** | 338.2 |
| ClipBERT [2] | 22.0 | 46.8 | 59.9 | 6.0 | - | - | - | - | - |
| Frozen [3] | 31.0 | 59.5 | 70.5 | 3.0 | - | - | - | - | - |
| Ours | **36.2** | **64.2** | 75.7 | 3.0 | **34.8** | **63.8** | 73.7 | **3.0** | **348.4** |

**Table 2:** Retrieval performance on the LSMDC dataset

| Method | Text-to-Video | | | | Video-to-Text | | | | SumR |
|---|---|---|---|---|---|---|---|---|---|
| | R@1 | R@5 | R@10 | MedR | R@1 | R@5 | R@10 | MedR | |
| JSFusion [31] | 9.1 | 21.2 | 34.1 | 27.0 | - | - | - | - | - |
| HiT [6] | 14.0 | 31.2 | 41.6 | 18.5 | - | - | - | - | - |
| TT-CE+ [28] | 17.2 | 36.5 | **46.3** | 13.7 | - | - | - | - | - |
| MEE [32] | 9.3 | 25.1 | 33.4 | 25.3 | - | - | - | - | - |
| MEE-COCO [32] | 10.1 | 25.6 | 34.6 | 21.0 | - | - | - | - | - |
| CE [8] | 11.2 | 26.9 | 34.8 | 25.3 | - | - | - | - | - |
| MMT [5] | 13.2 | 29.2 | 38.8 | 21.0 | 12.1 | 29.3 | 37.9 | 22.5 | 160.5 |
| DMM [7] | 15.8 | 34.1 | 43.6 | 14.3 | 14.3 | 33.7 | 43.6 | 15.5 | 185.1 |
| Frozen [3] | 15.0 | 30.8 | 39.8 | 20.0 | - | - | - | - | - |
| Ours | **17.8** | **37.1** | 45.9 | **13.5** | **16.5** | **34.3** | **44.6** | **14.0** | **196.2** |

# 4 Experiment

We evaluate our model's performance on several public datasets. In the sections that follow, we first introduce the datasets and metrics for comparing results, as well as implementation details. Then we demonstrating the overall performance of our model on the datasets. Finally, we present ablation studies to evaluate the effectiveness of each component of our model.

## 4.1 Datasets and Metrics

**MSR-VTT**

MSR-VTT [10] contains 10K YouTube videos, each of which is accompanied by 20 natural sentences that describe the video content. Following [3], we train on 9K train+val videos and report results on the 1K-A test set.

**MSVD**

MSVD [12] consists of 80K English descriptions for 1,970 videos from YouTube, with each video containing 40 sentences each. Following [3], we use the standard split of 1200, 100, and 670 videos for training, validation, and testing.

**DIDEMO**

DIDEMO [11] contains 10K Flickr videos annotated with 40K sentences. Following [3], we evaluate paragraph- to-video retrieval, where all sentence descriptions for a video are concatenated into a single query.

**LSMDC**

**Table 3:** Retrieval performance on DIDEMO

| Method | Text-to-Video | | | |
|---|---|---|---|---|
| | R@1 | R@5 | R@10 | MedR |
| S2VT [34] | 11.9 | 33.6 | - | 13.0 |
| FSE [35] | 13.9 | 36.0 | - | 11.0 |
| TT-CE+ [28] | 21.6 | 48.6 | 62.9 | 6.0 |
| CE [8] | 16.1 | 41.1 | - | 8.3 |
| ClipBERT [2] | 20.4 | 48.0 | 60.8 | 6.0 |
| Frozen [3] | 31.0 | 59.8 | 72.4 | 3.0 |
| Ours | **36.5** | **64.9** | **75.4** | **3.0** |

**Table 4:** Retrieval performance on MSVD

| Method | Text-to-Video | | | |
|---|---|---|---|---|
| | R@1 | R@5 | R@10 | MedR |
| VSE [36] | 12.3 | 30.1 | 42.3 | 14.0 |
| VSE++ [37] | 15.4 | 39.6 | 53.0 | 9.0 |
| TT-CE+ [28] | 25.4 | 56.9 | 71.3 | 4.0 |
| SUPP-SET [30] | 28.4 | 60.0 | 72.9 | 4.0 |
| MC [38] | 20.3 | 47.8 | 61.1 | 6.0 |
| CE [8] | 19.8 | 49.0 | 63.8 | 6.0 |
| Frozen [3] | 33.7 | 64.7 | 76.3 | **3.0** |
| Ours | **36.7** | **67.4** | **81.3** | **2.5** |

LSMDC [13] consists of 118,081 video clips sourced from 202 movies. The validation set contains 7,408 clips and evaluation is done on a test set of 1,000 videos from movies disjoint from the train and val sets. This follows the protocol outlined in [33].

**Metrics**

We evaluate retrieval performance using common information retrieval metrics such as Recall at K (R@K and K=1, 5, 10) and Median Rank (MedR). R@K is the proportion of test queries for which at least one relevant item is found among the top-K retrieved results. The MedR calculates the median rank of correct items in the returned ranking list, with a lower score indicating a better model. To reflect overall retrieval performance, we also take the sum of all R@K as SumR.

### 4.2 Implementation Details

We fintune the pretrained model from Frozen [3] on text-video retrieval dataset. The visual encoder have 12 attention blocks, patch size $P = 16$, sequence dimension $D = 768$, and 12 heads. The text encoder takes the architecture of DistilBERT base-uncased [39]. The dimensionality of the common text-video space is set to 256. We take some training settings same with Frozen [3]: we use 8 frames per video when training model; The temperature hyperparameter $\sigma$ for the loss defined in Equation (6) is set to 0.05; We randomly crop and horizontally flip during training and center crop the maximal square crop at test time for visual augmentation; All videos are resized to 224×224 as input; We use text augmentation during training for paragraph-retrieval settings, randomly sampling and concatenating a variable number of corresponding captions per video;The optimizer is Adam. We use a different learning rate scheduler: warmup for 5 epochs with a linearly growing learning rate from 0.0 to 0.00003, then use cosine decay scheduler to guide the learning rate from 0.00003 to 0.0 for other 45 epochs. The batch size is set as 64. The number of extra prototypes besides class token $K$ is set as 3. The variance loss parameter $\gamma, \epsilon$ are set as 0.75, $1e-4$. Variance loss weight $\alpha$ is set as 5.0.

### 4.3 Comparison to the State of the Art

Tables 1, 2, 3, 4 present the retrieval results of HiT on MSR-VTT, LSMDC, DIDEMO and MSVD. We also compare TMVM with other state-of-the-art methods: improving the robustness of pretrained model [23, 14, 24, 25, 26]; designing better architecture or loss [27, 28, 29, 9, 30, 6, 34, 35, 31] on pre-extracted features; focusing on mining information of different visual modalities [5, 7, 32]; end-to-end trainable model [2, 3].

As demonstrated by the results, TMVM outperforms all comparison methods clearly. We present video-to-text and text-to-video retrieval results for MSR-VTT and LSMDC. For MSR-VTT, our retrieval performance at SumR, in particular, is 348.4, outperforming recent state-of-the-art methods [7] by a margin of 10.2. For LSMDC, our retrieval performance at SumR also outperforms other methods by a clear margin. It accurately reflects TMVM's overall retrieval quality. We report retrieval performance in terms of text-to-video retrieval for DIDEMO and MSVD. TVMM continues to outperform comparison methods. Our method doesn't outperform R@10 of some other methods in Tables 1, 2 . It is possibly because the methods use multi modality information of video, such as speech, audio and appearance. Our method is based on Frozen [3] and only uses the frames.

**Table 5:** Ablation study on MSR-VTT to investigate the contributions of multiple visual prototypes matching and variance loss.

| Method | Multiple Prototypes $K$ | Variance Loss | Text-to-Video | | | |
|---|---|---|---|---|---|---|
| | | | R@1 | R@5 | R@10 | SumR |
| Baseline | 0 | ✗ | 34.1 | 61.0 | 72.8 | 167.9 |
| TVMM-Part | 3 | ✗ | 34.7 | 61.7 | 72.8 | 169.2 |
| TVMM-Decoder | 3 | ✗ | 35.1 | 61.8 | 74.0 | 170.9 |
| TVMM-Mask | 3 | ✗ | 35.9 | 63.3 | 74.5 | 173.2 |
| TVMM-Mask | 3 | ✓ | 36.2 | **64.2** | **75.7** | **176.1** |
| TVMM-Mask | 1 | ✓ | 35.4 | 63.1 | 73.5 | 172.0 |
| TVMM-Mask | 2 | ✓ | **36.4** | 64.1 | 75.1 | 175.6 |
| TVMM-Mask | 4 | ✓ | 34.8 | 63.5 | 73.4 | 173.1 |
| TVMM-Mask | 10 | ✓ | 33.7 | 62.7 | 72.5 | 168.9 |
| TVMM-Mask | $1569 = $ #tokens | ✓ | 32.3 | 59.8 | 71.7 | 163.8 |

**Table 6:** Ablation study on MSR-VTT to investigate the effect of the number of input video frames.

| Number of Video Frames | Method | Text-to-Video | | | |
|---|---|---|---|---|---|
| | | R@1 | R@5 | R@10 | SumR |
| 1 | Baseline | 24.6 | 48.5 | 60.2 | 133.3 |
| 1 | Ours | 25.3 | 50.8 | 61.2 | 137.3 |
| 4 | Baseline | 33.6 | 59.1 | 71.0 | 163.7 |
| 4 | Ours | 34.6 | 61.9 | 73.9 | 170.4 |
| 8 | Baseline | 34.1 | 61.0 | 72.8 | 167.9 |
| 8 | Ours | 36.4 | 64.2 | 75.7 | 176.1 |

## 4.4 Ablation Study

We discover that the proposed components, such as Multiple Visual Prototypes and Variance Loss, help to improve retrieval performance. We exhaustively and comprehensively ablate two components to demonstrate their effectiveness and robustness.

**Production of Multiple Visual Prototypes.** As mentioned above, we use mask-based method to aggregate token features from the final layer. In this section, we design two other variants to study whether this production is good enough.

- TVMM-Part. We divide the token features into several parts according to the frame order. The part's feature centers are then treated as prototypes.

- TVMM-Decoder. We use several embeddings as queries, and the token features as memory, then we throw query and memory into two-layer decoder to process the queries as visual prototypes.

Baseline is Frozen[3] with our training parameters. TVMM-Mask is our method which is introduced in Section 2. As shown in Table 5 ,TVMM-Mask outperforms TVMM-Part and TVMM-Decoder with a clear margin. TVMM-Part improved little compared with the baseline, only 1.3 on SumR. It shows that TVMM-Part is very limited in its ability to model multiple prototypes.

**The Number of Visual Prototypes.** As shown in Table 5, multiple prototypes is better than single prototype. With $K = 3$ and its variance loss, TVMM-Mask outperforms the baseline with 5.3 on RSum. This reflects a video can contain rich instance and event information and the positive text only describes a part of it. TVMM finds correspondence in the video with the help of multiple visual prototypes matching. The choice of number of prototypes is critical. K=3 is the best among 1,2,3,4 and 1569. When K is greater than or equal to 3, as K gets larger, the performance continues to degrade. A appropriate number of prototypes can assist the model in learning representative prototypes. When there are too many prototypes, the learned prototypes become noise rather than representative.

**The Number of Input Visual Frames.** In end-to-end training framework, the number of video frames is a parameter that has a big impact on performance. As shown in Table 6, as the number

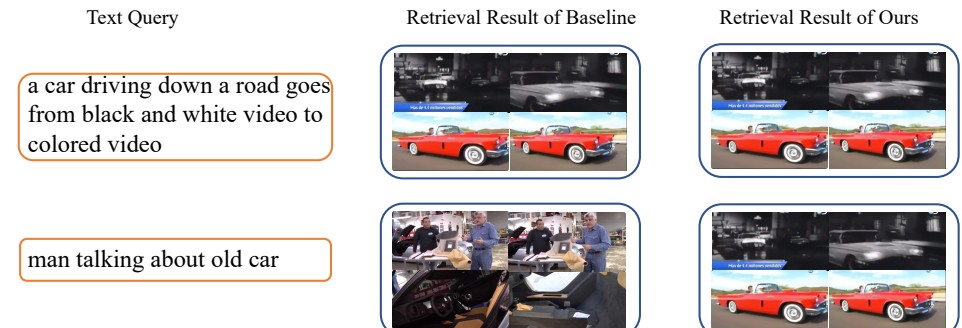

Text Query    Retrieval Result of Baseline   Retrieval Result of Ours

a car driving down a road goes from black and white video to colored video

man talking about old car

**Figure 4:** Some examples of text-video retrieval to compare the baseline with our method. The first video in the middle shows a guy talking about the transition from an old car to a new one. The second video in the middle shows a man describing various parts of a car to another person.

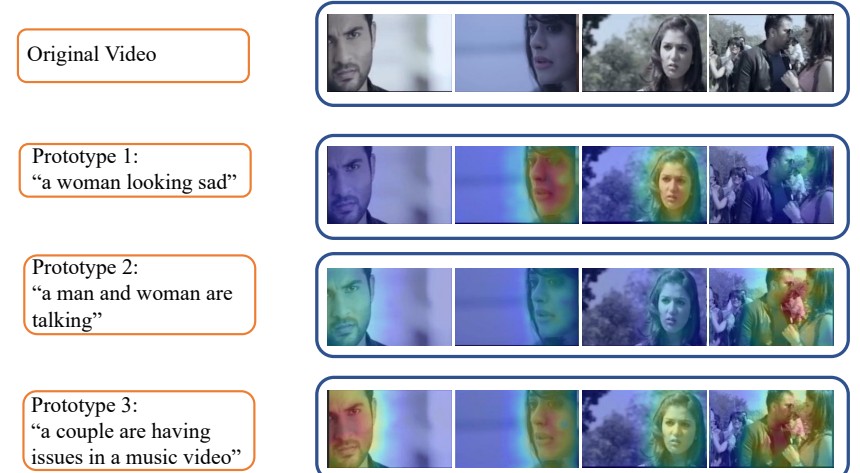

Original Video

Prototype 1: "a woman looking sad"

Prototype 2: "a man and woman are talking"

Prototype 3: "a couple are having issues in a music video"

**Figure 5:** Visualization of multi prototypes. We select a video and its three corresponding captions from MSR-VTT. The captions are matched with the prototype that was the most similar. Then, to visualize the prototypes, we plot the heatmaps based on the mask values on the four sampled frames. The four frames depict a serious man, a sad woman, a serious woman, and a pair talking to each other.

of video frames increases, so does the performance. It's worth noting that even on single-frame input, our method also improves the baseline with 4.0 on SumR. It proves the importance of multiple prototype matching.

**Some Retrieval Examples.** Each test video in the original MSR-VTT split has only one text description. In this experiment, we take some samples from the training data set and add them to the test set to demonstrate the superiority of the method. When compared to the baseline, our method provides correct retrieval results for the two test text queries, as shown in Figure 4. It is because our TVMM model can alleviate the ambiguity problem in video-text correspondence.

**Visualization of Multi Prototypes.** We choose a video and its three captions from MSR-VTT. The captions are matched with the most similar prototype. The heatmaps based on the mask values on the four sampled frames are then plotted to visualize the prototypes. The four frames show a serious man, a sad woman, a serious woman, and a pair conversing. As you see in Figure 5, the three captions correspond well with the prototypes, which attend to different contents of the video.

## 5 Conclusion

In this paper, we have proposed a Text-Adaptive Multiple Visual Prototype Matching Model. It generates multiple prototypes for video automatically to account for its rich information, and we present a text-adaptive matching strategy to find correspondence between texts and videos. Then, we propose a variance loss to encourage different prototypes to pay attention to different video contents.

Experimental results on four public video retrieval datasets demonstrate the advantages of our model. Ablation studies are also carried out to verify the effectiveness of each part of our model. Our study has one drawback in that the number of prototypes is specified manually rather than automatically. Typically, the number of prototypes in each video varies. That is our future work.

## 6   Acknowledgement

This work was supported by the National Science Foundation for Young Scientists of China (62106288). And it was in part supported by Foshan HKUST Projects (FSUST21-FYTRI01A, FSUST21-FYTRI02A). C. Shen's participation was in part supported by a major grant from Zhejiang Provincial Government.

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
