# OpenReview forum: "Text-Adaptive Multiple Visual Prototype Matching for Video-Text Retrieval"
_NeurIPS.cc/2022/Conference — NeurIPS 2022 Accept_

### Official Review · Reviewer_Xjvw · 2022-07-08

**Rating:** 6
**Confidence:** 5
**Soundness:** 2 fair
**Presentation:** 3 good
**Contribution:** 2 fair

**Summary:**

This paper tackles the task of video-text retrieval. Motivated by the fact that multiple different text descriptions can apply to the same video the paper proposes a new model where the video is represented by multiple prototypes in the video-text embedding space. This allows multiple different texts to have a high similarity score with the video. To encourage these different texts to be diverse the paper also proposes a variance loss. The proposed method is tested on 4 video-text retrieval datasets: MSR-VTT, MSVD, DIDEMO and LSMDC.


**Questions:**

- What are the advantages of the method proposed in this paper over the methods proposed in [A] and [C]?
- Does this paper still achieve state-of-the-art results when compared to the results published in [C]?
- Why were Didemo and LSMDC chosen as datasets to evaluate on over VATEX or ActivityNet Captions?
- Why does the proposed method improve on R@1 and R@5 but not on R@10 in Tables 1 and 2?
- Does not having exhaustive annotations of video-text relevancy cause any issues with the evaluation?

**Limitations:**

Limitations or ethical concerns are not mentioned which is a shame considering there was a half page of space left which could have been used for this.

**Strengths And Weaknesses:**

## Strengths
***
- The observation that multiple different text descriptions can apply to the same video is interesting.
***
- The analysis on inter-text and intra-text similarity of captions for different videos is valuable.
***
- The proposed method is simple and the ablations should a clear improvement over the baseline approach.
***

## Weaknesses
***
- the main issue I have with this paper is that it ignores prior works which consider multiple relevant captions for images/videos
- [A] considers one-to-many correspondences between images and text and proposes a probabilistic embedding method so that multiple texts can be matched with a single image. The related work of this paper also highlights previous methods in images considering many-to-many correspondences which are also relevant.
- In video [B] highlighted issues with the one-to-one assumption in evaluation of video retrieval methods and proposes a new evaluation metric to address this. It also performs some additional experiments where videos are trained with other relevant captions from the training set in addition to their annotated captions.
- [C] considers multi-query video retrieval, where multiple queries are used in both training and testing. [C] also has an experiment where they show multi-query training can improve on single-query testing (Figure 6) and the results seem to out-perform the ones in this paper
- Without these works being acknowledged in related work and the differences betweeen these and the newly proposed method being explained I don't think this paper can be accepted.
- Where appropriate prior works which consider multiple relevancies should also be compared to. This paper claims state of the art but it appears [C] may actually outperform it. I would like clarification from the authors in the rebuttal on this point.
- I also think [A] should be compared to. It was created for image-text retrieval but since the motivation and problem it tackles are so similar to this paper it seems like a valuable comparison, particularly as the proposed method in this paper doesn't seem to be specific to video other than the backbone network the features come from.
***
- the choice of datasets is strange
- MSR-VTT and MSVD make sense as each video is annotated with multiple captions
- However, Didemo and LSMDC don't seem to fit with the problem tackled in the paper. For Didemo the paper states 'all sentence descriptions for a video are concatenated into a single query' and for LSMDC there is no mention of there being multiple captions per video.
- VATEX and ActivityNet Captions would be suitable as they both have multiple captions per video
***
- Little insight is iven about the results
- for instance, the proposed method consistently improves on Recall@1 and Recall@5 but doesn't on Recall@10 (Tables 1 and 2). This is not explained.
***
- There is a potential issue with the annotations for the datasets used as the datasets are not exhaustively annotated with possible captions
- This means it is unknown whether captions from another video may correctly describe part of the video in question. Therefore, some correct retrievals may be penalized by the evaluation metric.
- This not reason to reject this paper as this is potentially also a problem with prior works in video-text retrieval. I'm just interested in whether the authors think this is an issue and how much it would effect the results.
***
[A] Probabilistic Embeddings for Cross-Modal Retrieval. Sanghyuk Chun, Seong Joon Oh, Rafael Sampaio de Rezende, Yannis Kalantidis, Diane Larlus. CVPR 2021.
[B] On Semantic Similarity in Video Retrieval. Michael Wray, Hazel Doughty, Dima Damen. CVPR 2021.
[C] Multi-query Video Retrieval. Zeyu Wang, Yu Wu, Karthik Narasimhan, Olga Russakovsky. ArXiv, Jan 2022.

## Final Justification
My main concern prior to the rebuttal was that the paper ignore prior works which shared the motivation of multiple different text descriptions being applicable to a video or image. The rebuttal has addressed this concern by providing comparison to [A], an image-focused method which uses a probabilistic embedding, and [C] a video retrieval method which uses multiple relevant captions in training. The revised version of the paper also doe include a little more insight into the results, although this could have been better. I do appreciate the new Figure 5 visualizing the mask values and prototypes. With the new results comparing the paper to similar prior works I recommend accepting this paper, however I think it is important that in the final version of the paper these prior works are also included in the related work and the difference with this work is clearly highlighted.

---

> ### Author Response · Authors · 2022-08-01
> **Response**
>
> ###  1. Comparison of our method with [A], [B] and [C]
> Thanks for the suggestions. In comparison to [A], our method incorporates a novel variance loss (Section 3.3) and an efficient matching function (Section 3.2) for learning multiple prototype embeddings.  Compared with image,  video contains much more instances and events.  So the corresponding ambiguity problem in Video-Text Retrieval is more serious than Image-Text Retrieval. The variance loss helps reduce the ambiguity by encouraging different prototypes to attend to different contents of video. Using the same baseline, we implemented [A] with the same K. Our matching method without variance loss outperforms [A] by 4.8 points on SumR on MSR-VTT dataset, indicating its superiority.
>
> | Method        | R@1           | R@5  |  R@10           | SumR |
> | ------------- |:-------------:| -----:| -----:| -----:|
> | Ours matching + variance loss       |  36.2      |  64.2  | 75.7  | 176.1 |
> | Ours matching      |  35.9      |  63.3  | 74.5  | 173.2 |
> |  [A] matching     | 34.3 | 61.5 | 72.6 | 168.4 |
>
> Ours is very different from B. First, the research problems are different. We focus on  that video contents can be diverse, and the text  descriptions can be quite different. [B] emphasizes that the captions collected for each video are not unrelated to other videos.
> Second, the methods are different.  Our method generates multiple representative visual prototypes to match text feature. [B] estimates the semantic similarities of all  text-video  pairs by Bag-of-Words Semantic Similarity, Part-of-Speech Semantic Similarity and METEOR Similarity, etc.
>
>
> The method and experiment setting in [C] is under a different setting with ours. In [C], model is evaluated under multi-query setting (Section 3.3 in [C]), which is significantly easier.  It searches the video archive using multiple descriptions, whereas we only use one text query to retrieve relevant videos.
>
> We will add these comparisons to the revised version.
> ### 2. The usage of  Didemo and LSMDC
> Our work is based on Frozen paper  "Frozen in Time: A Joint Video and Image Encoder for End-to-End Retrieval". So we follow it to use the two datasets.  And the Video-Text Correspondence Ambiguity problem also exists in the two datasets. We  demonstrate that, even without multiple captions for each video, our model will automatically find multiple prototypes and improve the performance.
>  	The Video-Text Correspondence Ambiguity problem also exists in the two datasets. We  want to demostrate that, even without multiple captions for each video, our model will automatically find multiple prototypes and improve the performance.
>
> ### 3. The performance comparison
> Our method does improve significantly upon Frozen baseline on R@10 metric on multiple datasets. Our method doesn't  outperform R@10 of some other methods on some datasets. It is  possibly because  the methods use multi modality information  of video, such as speech, audio and appearance. Our method is based on Frozen and only uses the frames.
>
> We have  added this analysis in Section 4.3 in the revised version. And we will integrate multi-modal information of video in feature work.
> ### 4. The annotation problem
> The  lack of exhaustive annotations of video-text relevancy  is a problem. It will improve model learning and provide a better evaluation metric. We will do it in future work.

---

> > ### Comment · Reviewer_Xjvw · 2022-08-05
> > **Comparison with [C]**
> >
> > I'd like to thank the authors for responding to my initial points and the new comparison to [A], however I still do not understand why [C] cannot be compared to. I understand that [C] proposes to train and test a video-text retrieval model with multiple queries, however it also provides results when training with multiple queries and testing with a single query. These results are provided in [C], table 2 with the R@1_1 metric and discussed in section 4.3 '(R4) Evaluation with varying number of queries', where the conclusion is that 'the de-noising effect of multi-query training can also be utilized to improve standard single-query retrieval'.
> >
> > Could the authors please explain why these results in [C] are not comparable?

---

> > > ### Author Response · Authors · 2022-08-07
> > > **Comparison with [C]**
> > >
> > > Thanks to the suggestion. We have added  the experiment of  our method based on [C]'s strong baseline in single-query retrieval setting. Our method significantly outperforms [C] by 1.2 point in R@1 on the MSRVTT dataset, as shown in the table below, indicating its superiority.
> > > | Method	| R@1 |
> > > |-----------------|----------|
> > > |    [C]'s Baseline	| 41.5 |
> > > |         [C]         | 41.9  |
> > > | [C]'s Baseline + Our Method	 | 43.1 |
> > >
> > > The experiment is based on the code released by [C] at https://github.com/princetonvisualai/MQVR. We will add this to the revised version.

---

> > > > ### Author Response · Authors · 2022-08-09
> > > > **Gentle Reminder**
> > > >
> > > > Dear Reviewer Xjvw,
> > > > Thank you very much again for the time and effort put into reviewing our paper. We believe that we have addressed all your concerns in our response. We have also followed your suggestion to improve our paper and have added additional experimental results. We kindly remind you that we are approaching the end of the discussion period. We would love to know if there is any further concern, additional experiments, suggestions, or feedback, as we hope to have a chance to reply before the discussion phase ends.

---

> > > > > ### Comment · Reviewer_Xjvw · 2022-08-09
> > > > > **Questions Answered**
> > > > >
> > > > > Thank you for the additional response, I don't have any further questions.

---

### Official Review · Reviewer_UJze · 2022-07-08

**Rating:** 7
**Confidence:** 4
**Soundness:** 4 excellent
**Presentation:** 3 good
**Contribution:** 3 good

**Summary:**

When performing video retrieval from a text query (or vice versa), representing each video with a single embedding can be problematic because a video can depict many things, and potentially match multiple different search queries. One solution is to represent the video by multiple embedding "prototypes", and define video-text similarity to be the similarity between a text query and the nearest of the video's prototypes.

The authors propose to create K different prototypes for each video by computing ViT tokens for the ~8 input frames, and applying K-different learned functions, each of which masks out a subset of the tokens and sums the remaining ones. These functions are learned as part of dual text-video encoder setup using symmetrical contrastive loss.

Of course if the learning model produces several nearly-identical prototypes for a video, then there's no gain. To prevent this, a training loss term is introduced to maximize the variance of the aforementioned mask functions, encouraging diversity between prototypes.

Strong recall numbers are demonstrated on a variety of video-text retrieval datasets. The components of the model are validated with ablation studies.


**Questions:**

Line 38: Why exactly is it infeasible to align a single video feature with multiple text features? It might be worth elaborating here.

Line 158: "all other pairwise combinations in the batch are deemed negative". Of course, this means that even if a video has 20 potential ground truth queries (e.g. MSR-VTT) it'll only appear in a batch once, correct? It might be worth clarifying this in the paper.

Are the mask values interpretable? For example, for a given video, could we see that mask#1 focuses on ViT tokens representing the woman in the video, mask#2 focuses on the man, and mask#3 on the crowd? This could make a compelling visualization.


**Limitations:**

The authors did not describe any shortcomings. I would appreciate some mention of where the model fails or could be improved. For example, the premise that a query need only match a single prototype might lead to low-precision results (think about matching based on an OR vs. AND operator). Is this observed? Are there any other areas for future work?

I don't see any significant potential negative societal impact.



**Strengths And Weaknesses:**

**Originality:** Although the idea of representing a complex item with multiple prototypes for the purposes of retrieval is not novel, this particular instantiation -- learning K different views that aggregate ViT tokens across all the input frames -- is novel and very interesting, particularly because it's learned rather than ad-hoc, and can potentially attend to one element across many frames of the video.

I would like to have seen a little more investigation as to what the model is learning, e.g. a visualization of what each mask is attending to, or a demonstration that all of the prototypes for a video are useful -- matching different ground truth queries.

**Quality:** The method and evaluation seem sound and convincing. It might have been nice to see a comparison to a baseline with one prototype per frame, as proposed on lines 122-123.

**Clarity:**

Line 130: What does "we find that the mask-based production of kth is enough" mean?

What baseline method is used in Tables 5 & 6? I couldn't find it explained anywhere.

Equation (2). What is the dimensionality of the output of f^mask? Is it 1-dimensional or D-dimensional?

The paper would benefit from additional editing for grammar.
Some references are incorrect, e.g. "Figure 3.3" on line 118.
There are duplicate sentences on lines 105 and 107.

**Significance:** I think the idea is technically novel, and the performance on standard benchmarks is strong enough that this paper would be of interest to the video search/retrieval community, and possibly to the video-text representation learning community as well.

---

> ### Author Response · Authors · 2022-08-01
> **Response**
>
> ### 1. The alignment of single video feature and multiple text features.
> Thanks for the suggestion.  We have changed it in L38 in the revised paper as
> " The text descriptions are very diverse and scattered in the feature space. With diverse video contents , it is infeasible to align a single video feature with the diverse text features."
>
> ### 2.  A video appears at most once in a batch?
> Yes, it is. Thanks for the suggestion. We have added a sentence "Each video appears at most once in each batch to prevent duplicates" in Section 3.4 in the revised paper.
>
> ###  3. Visualization of mask values
> Thanks for the suggestion. We have added Figure 5 to visualize the mask values and prototypes in the revised version.  We choose a video and its three captions from MSR-VTT. The captions are matched with the most similar prototype. As show in Figure 5 in the revised paper, mask values on the four sampled frames   are used to plot the heatmaps to visualize the prototypes. The four frames show a serious man, a sad woman, a serious woman, and a pair conversing. As you see, the three captions correspond well with the prototypes, which attend to different contents of the video.
>
> ### 4. What does "we find that the mask-based production of kth is enough" mean?
> Sorry for the confusion by our grammer error. We have changed it as "However, we find that the mask-based production of the prototypes is good enough.", following "There are many methods to aggregate the token features to produce multiple visual prototypes, such as transformer decoder, graph neural network, LSTM etc. "  in the revised paper.
>
> ### 5. What is the baseline?
> Sorry for the confusion. The baseline is Frozen with our training parameters. We haved added this definition in Section 4.4. The detailes of training parameters are in Section 4.2.
>
> ### 6. The dimensionality of the output of f^mask
> It is K, the number of prototypes.
>
> ### 7. Some Grammer Mistakes
> Thanks for the suggestions. We have fixed the grammer mistakes in the revised paper.

---

### Official Review · Reviewer_DonZ · 2022-07-11

**Rating:** 5
**Confidence:** 4
**Soundness:** 3 good
**Presentation:** 3 good
**Contribution:** 3 good

**Summary:**

The paper tackles the problem of text-video retrieval. The authors argue that current models concentrate on mapping one video to one sentence using only one feature that needs to be matched with multiple texts is hard to achieve and instead propose a text adaptive multiple visual prototype matching where the model captures multiple prototypes that describe a video. In this way, the query is matched with the most similar prototype instead of having only one video level representation. Finally, the authors test their approach on four retrieval benchmarks.

**Questions:**

1. Is the model initialized with Frozen?

2. Have you tried the idea in conjunction with other methods?

3. What do the groups correspond to in the Tables?

4. What happens if K is larger? Any intuition on why there seems to be a decrease in performance with a higher K.

5. Does the Baseline in Table 5 correspond to Frozen?

**Limitations:**

The limitations are somehow discussed through the ablations. The potential societal impact is not discussed.

**Strengths And Weaknesses:**

The paper tackles an important problem text-video retrieval and has a interesting idea at its basis. It reports results on four datasets and has good results.

My main concern is related to the novelty. While I like the idea and find it interesting, I fell like it is quite closely related to the idea of probabilistic embeddings that was introduced for text-image retrieval. While there are definitely differences, I feel like a proper comparison and discussion between the two needs to be added.

* Chun, Sanghyuk, et al. "Probabilistic embeddings for cross-modal retrieval." Proceedings of the IEEE/CVF Conference on Computer Vision and Pattern Recognition. 2021.

Figure 4 - maybe make the picture larger (there still is space on the sides, so the frames can be made larger)

Minor:
lines 169-170 look weird - can they be moved to the next page?

---

> ### Author Response · Authors · 2022-08-01
> **Response**
>
> ### 1. Comparison with "Probabilistic embeddings for cross-modal retrieval"
> Thanks for the suggestion. In comparison to PCME("Probabilistic embeddings for cross-modal retrieval"), our method incorporates a novel variance loss (Section 3.3) and an efficient matching function (Section 3.2) for learning multiple prototype embeddings.  Compared with image,  video contains much more instances and events.  So the corresponding ambiguity problem in Video-Text Retrieval is more serious than Image-Text Retrieval. The variance loss helps reduce the ambiguity by encouraging different prototypes to attend to different contents of video. Using the same baseline, we implemented PCME with the same K. Our matching method without variance loss outperforms PCME by 4.8 points on SumR on MSR-VTT dataset, indicating its superiority.
>
> | Method        | R@1           | R@5  |  R@10           | SumR |
> | ------------- |:-------------:| -----:| -----:| -----:|
> | Ours matching + variance loss       |  36.2      |  64.2  | 75.7  | 176.1 |
> | Ours matching      |  35.9      |  63.3  | 74.5  | 173.2 |
> | PCME matching     | 34.3 | 61.5 | 72.6 | 168.4 |
>
> We will add this to the revised paper
> ### 2. Is the model initialized with Frozen?
> Yes. Our model is initialized with Frozen.
> ### 3. Conjunction with other Methods
> Thanks for the suggestions. We have not combined our prototype idea with other methods but will do so in  future work.
> ### 4. Group of Table
> As shown in Section 4.3,  the four groups are: improving the robustness of pretrained model; designing better architecture or loss based on  pre-extracted features; focusing on mining information of different visual modalities; end-to-end trainable model.
> ###  5. The results of Larger K
> When K is greater than or equal to 3, as K gets larger, the performance continues to degrade. When K=#tokens ,  the SumR on the MSR-VTT dataset is 163.8, which is significantly worse than our proposed setting.  An appropriate number of prototypes can assist the model in learning representative prototypes.  When there are too many prototypes, the learned prototypes tend to become noise rather than representative.
> |     K   | SumR |
> |--------|-------|
> | 2    | 175.6    |
> | 3   | 176.1    |
> | 4 | 173.1 |
> | 10 | 168.9 |
> | 1569=#tokens | 163.8 |
> We have added this experiment  in Table 5 in the revised paper.
>
> ### 6. Baseline is Frozen?
> Sorry for the confusion. The baseline is Frozen with our training parameters. We haved added this definition in Section 4.4. The detailes of training parameters are in Section 4.2.

---

> > ### Comment · Reviewer_DonZ · 2022-08-07
> > **Thank you for the response**
> >
> > Hi,
> >
> > The provided response answer my questions. Thanks!

---

### Official Review · Reviewer_nbMi · 2022-07-11

**Rating:** 5
**Confidence:** 5
**Soundness:** 3 good
**Presentation:** 3 good
**Contribution:** 3 good

**Summary:**

The paper studies the video and text retrieval task. As the clip/video contains rich information in a sequence of frames, while the text contains limited information about one clip/video. There is a semantic gap between the video and text matching. To bridge the semantic gap, the author proposed to learn prototypes from the video and match the text with the video via those learnt prototypes. The prototypes should capture different semantic level information about one video. To learn these prototypes, the author proposed text-adaptive multiple visual prototype matching model. Specifically, it maps a video into a set of visual representation. It then learns an aggregation function to aggregate those visual representation to form the prototypes. Then the prototypes are used to match the text for video and text retrieval. The proposed approach achieves better performance on several video and text retrieval datasets.

**Questions:**

Please address the Q1, Q2, Q3, and Q4. My main concern is that the prototype might just be an interesting story for the proposed approach. It might not be the main reason for the improved performance.

**Limitations:**

There is no negative societal impact.

**Strengths And Weaknesses:**

Pros:
The main argument that there exists a semantic gap between the text and video is important and interesting. The proposed model that using the prototypes to represent video and matching the prototypes with text is also interesting and promising. Although how to define the prototype is an open question, this approach provides one of the definition of the prototype and also shows one implementation of this idea.

Cons:
1. Prototypes: I wonder whether the prototypes really capture the semantic meaning of the video. Concretely, what is the performance of setting the # of prototypes as the same as the # of tokens (B as in L128)? I wonder the improved performance is due to the similarity function (Eq. 3) or due to the proposed prototypes?

2. I didn't fully understand the variance loss. Specifically, L145-L146 'we'd like all masks in a batch of videos ...' is pretty confusing.

3. What is the meaning of the Figure 3? Specifically, what is the definition of the 'minimum of intra-text similarity'? And how to interpret the Figure 3? Although L112-L121 tries to explain this figure, I still didn't understand the aim and the meaning of this analysis.

4. As the aims of prototypes is to capture different semantic meanings of video, I wonder does the proposed prototypes really achieve this goal? Specifically, is there any qualitative or quantitative results/experiments to verify this argument?

5. My main concern is the improved performance might not due to the proposed prototypes.

---

> ### Author Response · Authors · 2022-08-01
> **Response**
>
> ### 1. The Number of Prototypes
> We have added experiments with #tokens as K in Table 5 in the revised paper. The SumR on the MSR-VTT dataset is 163.8, which is significantly worse than our proposed setting of 176.1. This  suggests that a smaller K is important for better performance and the similarity  function is not the primary reason for the improvement.
>
> ### 2. Clarification of Variance Loss
> Sorry for the confusion. As shown in Figure 3, some videos have multiple events with diverse text descriptions, while others only have one event with very similar text descriptions.  So we aim to enforce the masks of different prototypes of all videos to be diverse. The variance loss forces the variance of all prototype mask values for each token to be large. We have clarified this  in Section 3.3 in the revised version.
>
> ###  3. Clarification of Figure 3
> Sorry for the confusion. In the dataset of MSR-VTT, each video is paired with 20 natural sentences.  As shown in L115, intra-text similarity is defined as  same-video text similarity  of MSR-VTT and inter-text similarity is defined as different-video text similarity of MSR-VTT. As a result, the diversity of video contents  can be represented by the minimum intra-text similarity.  As shown in Figure 3,   At least one-third of the minimum of intra-text similarity is smaller than the mean of inter-text similarity on features from both models.  It demonstrates that many videos have diverse captions and contents and it is necessary to use multiple visual prototypes to align with different text descriptions. We have clarified this in Figure 3 and Section 3.2 in the revised version.
>
> ### 4. Visualization of Prototypes
> Thanks for the suggestion. We have added Figure 5 to visualize the prototypes in the revised paper.  We choose a video and its three captions from MSR-VTT. The captions are matched with the most similar prototype. As shown in Figure 5 in the revised paper,  The heatmaps based on the mask values on the four sampled frames are then plotted to visualize the prototypes. The four frames show a serious man, a sad woman, a serious woman, and a pair conversing. As you see, the three captions correspond well with the prototypes, which attend to different contents of the video.

---

> > ### Comment · Reviewer_nbMi · 2022-08-10
> > **Post rebuttal comments.**
> >
> > Thanks authors for the comments. The comments resolved my questions. I maintain the ratings.

---

### Meta-Review · Area_Chair_y6mL · 2022-08-21

**Recommendation:** Accept
**Confidence:** Certain

**Metareview:**

For the one video vs multiple descriptions issue in video-text retrieval, this paper presents a text-adaptive multiple visual prototype matching model, which could adaptively find the suitable visual prototype for an arbitrary textual description. And comprehensive evaluation results were conducted. The rebuttal successfully addressed some of the major concerns and, in the end, there is a general consensus about accepting the paper. Moreover, most reviewers suggest that the authors update the Related Work section with a more thorough comparison with the papers mentioned by reviewer Xjvw.

**Award:**

No

---

### Decision · Program_Chairs · 2022-09-14

Accept